# HIF-1α-Overexpressing Mesenchymal Stem Cells Attenuate Colitis by Regulating M1-like Macrophages Polarization toward M2-like Macrophages

**DOI:** 10.3390/biomedicines11030825

**Published:** 2023-03-08

**Authors:** Wenya Zhu, Qianqian Chen, Yi Li, Jun Wan, Jia Li, Shuai Tang

**Affiliations:** 1Medical School of Chinese PLA, Beijing 100039, China; 2Department of Geriatrics, The Sixth Medical Center, Chinese PLA General Hospital, Beijing 100048, China; 3Department of Gastroenterology, The Second Medical Center & National Clinical Research Center for Geriatric Diseases, Chinese PLA General Hospital, Beijing 100039, China; 4Department of Gastroenterology, The First Medical Center, Chinese PLA General Hospital, Beijing 100039, China

**Keywords:** mesenchymal stem cells, hypoxia-inducible factor1α, colitis, immunoregulation, macrophages, phosphatidylinositol 3-kinase-γ

## Abstract

A modified mesenchymal stem cell (MSC) transplantation is a highly effective and precise treatment for inflammatory bowel disease (IBD), with a significant curative effect. Thus, we aim to examine the efficacy of hypoxia-inducible factor (HIF)–1α-overexpressing MSC (HIF-MSC) transplantation in experimental colitis and investigate the immunity regulation mechanisms of HIF-MSC through macrophages. A chronic experimental colitis mouse model was established using 2,4,6-trinitrobenzene sulfonic acid. HIF-MSC transplantation significantly attenuated colitis in weight loss rate, disease activity index (DAI), colon length, and pathology score and effectively rebuilt the local and systemic immune balance. Macrophage depletion significantly impaired the benefits of HIF-MSCs on mice with colitis. Immunofluorescence analysis revealed that HIF-MSCs significantly decreased the number of M1-like macrophages and increased the number of M2-like macrophages in colon tissues. In vitro, co-culturing with HIF-MSCs significantly decreased the expression of pro-inflammatory factors, C-C chemokine receptor 7 (CCR-7), and inducible nitric oxide synthase (INOS) and increased the expression of anti-inflammatory factors and arginase I (Arg-1) in induced M1-like macrophages. Flow cytometry revealed that co-culturing with HIF-MSCs led to a decrease in the proportions of M1-like macrophages and an increase in that of M2-like macrophages. HIF-MSCs treatment notably upregulated the expression of downstream molecular targets of phosphatidylinositol 3-kinase-γ (PI3K-γ), including HIF-1α and p-AKT/AKT in the colon tissue. A selected PI3K-γ inhibitor, IPI549, attenuated these effects, as well as the effect on M2-like macrophage polarization and inflammatory cytokines in colitis mice. In vitro, HIF-MSCs notably upregulated the expression of C/EBPβ and AKT1/AKT2, and PI3K-γ inhibition blocked this effect. Modified MSCs stably overexpressed HIF-1α, which effectively regulated macrophage polarization through PI3K-γ. HIF-MSC transplantation may be a potentially effective precision therapy for IBD.

## 1. Introduction

Inflammatory bowel disease (IBD), including Crohn’s disease and ulcerative colitis, is a progressive, immune-mediated, chronic, and recurrent inflammatory disease with an undetermined etiology and a rapidly increasing global prevalence [1,2]. New research reveals that immunomodulatory therapy targeting inflammatory factors and immune cells shows promising better results in IBD therapy [3]. Mesenchymal stem cells (MSCs) are multipotent cells characterized by self-replication ability, multilineage differentiation, and specific surface markers [4], having properties in tissue regeneration and immunomodulation [5]. MSCs have been shown to effectively suppress inflammation and have immunoregulatory effects in IBD, albeit challenged by cost and the incomplete characterization of the effect [6]. Genetic modification is one strategy for optimizing MSC function, including homing and immunoregulation [7]. Transplantation with optimized stem cells should be considered as a method for the precise treatment of IBD. For example, pretreatment with IFN-γ and pre-exposure to muramyl dipeptide potentiate the effect of MSCs [8,9]. Since MSCs are considered to originate in hypoxic niches such as bone marrow [10], they may stably express HIF-1α in normoxic environments [11], thereby adapting to hypoxic environments by inducing glycolysis and contributing to the maintenance of an undifferentiated state [12]. Hypoxia preconditioning effectively optimizes the immunoregulatory effect of MSCs [13]. Hypoxia-inducible factor (HIF) plays the most important role in regulating the hypoxia response [14]. HIF-1α is a mediator of cellular adaptation to hypoxia and generally regulates the metabolism and multipotency of MSC. Moreover, intestinal tracts have a steep oxygen gradient under physiological conditions, while the inflamed intestinal mucosa presents as severely hypoxic. Oxygen-sensitive prolyl hydroxylases (PHDs), critical regulators of HIF-1α, have been identified as promising therapeutic targets in IBD [15]. Therefore, we hypothesized that the stable overexpression of HIF-1α in MSCs will improve cellular adaptation to the hypoxic intestinal environment of IBD, thereby enhancing the function of MSCs in inflammatory intestinal tracts, including immunoregulation.

Immuno-therapeutic mechanisms of MSCs include regulating inflammatory factors, controlling key molecules of inflammatory pathways, rebuilding the lymphocyte subtype balance (Th1/Th2 and Th17/Treg), inducing T-cell apoptosis, and polarizing macrophages [16,17]. The mechanisms of MSCs regulating the metabolic state and polarizing macrophages include cell–cell contact, extracellular vesicles, and immunosuppressive factors [18]. Macrophages play a key role in the initiation and resolution of inflammation, with dynamic polarization processes between M1 and M2 subtypes [19]. The largest population of macrophages resides in the gastrointestinal tract, and Lissner et al. [20] showed that M1 macrophages induce intestinal barrier disruptions by downregulating tight junction proteins and inducing epithelial cell apoptosis. The plasticity of macrophages during the gut inflammatory response and the cascade regulation effect of inflammatory factors, such as IL-23 and IL-10, on other immune cells indicate that macrophages are potential targets of therapeutic intervention in IBD [21,22]. Moreover, alternatively activated macrophage (M2 macrophage) differentiation is a successful IBD therapy [23]. Based on these findings, we attempted to determine whether optimized MSCs maintain the intestinal immune balance and alleviate colitis by regulating macrophage polarization.

This study aims to elucidate the potential effects and mechanisms of HIF-1α-overexpressing MSCs (HIF-MSCs) in modulating the immune balance and disease relief in IBD by targeting macrophage polarization and the potential key role of PI3K-γ in the mechanism, which has not been reported yet. HIF-1α overexpression was achieved via lentiviral transfection of MSCs isolated from the human umbilical cord. We transplanted 2,4,6-trinitrobenzene sulfonic acid (TNBS)-induced experimental colitis mice, which we had previously established as a successful chronic colitis mouse model, with HIF-MSCs.

## 2. Materials and Methods

### 2.1. MSC Culture

Human umbilical cord MSCs (hUCMSCs) were isolated from the umbilical cord tissue of volunteers, who provided informed consent. All samples were collected after a normal delivery at the Obstetrics Department of the Sixth Medical Center, Chinese PLA General Hospital. Primary cells were obtained with Wharton’s jelly using the explant method [24]. Wharton’s jelly was separated from the endothelial tissue and cut into pieces of 1–2 mm. The pieces were then mixed with minimum essential medium (Gibco, St. Emeryville, CA, USA) containing 10% fetal bovine serum (FBS; Gibco, St. Emeryville, CA, USA) and cultured at 37 °C in the presence of 5% CO_2_ for 7–14 days; the medium was replenished every 3 days. The non-adherent tissues and suspended blood cells were removed by medium replacement. Single adherent cells crawled out of the edge of Wharton’s jelly tissue at days 4–5, and cell clusters were gradually formed. The density of the adherent cells reached up to 70–80% at days 10–12. The adherent cells were recorded as P0 generation and passaged in the ratio of 1:2. Primary cells were passaged to the P3 generation for subsequent experiments. All the above operations were performed under aseptic conditions.

### 2.2. Recombinant Lentivirus Design and Transfection

The backbone vector, Ubi-MCS-3FLAG-CBh-gcGFP-IRES-puromycin (Hanbio, Inc., Shanghai, China), was used to reconstruct a lentiviral vector expressing HIF-1α, fragments of which were amplified with the primers listed in Appendix A. The plasmid was extracted from the correctly sequenced bacterial solution with a Plasmid Mini Kit (TIANGEN BIOTECH, Beijing, China). In the logarithmic growth stage, 293T cells were transfected with the plasmid, and the supernatant was collected to obtain a purified and concentrated vector.

At passage three, MSCs at a density of 2 × 10^5^ cells/mL were inoculated and cultured for 16–24 h until the cell confluence was 30–40%. Green fluorescent protein (GFP)-labeled HIF-1α-expressing lentivirus with puromycin resistance was diluted with complete medium to a titer of 1 × 10^8^ transduction units (TU)/mL and then added to the medium at a multiplicity of infection of 20. Then, it was allowed to cultivate for 18 h after infection, followed by refreshment of the complete medium. The cell morphology was normal during the entire process. Exactly 1 µg/mL of puromycin (Sigma, St. Louis, MO, USA) was added to the culture after 48 h to eliminate the uninfected MSCs. The GFP fluorescent signal of successfully infected cells (HIF-MSCs) was observed using a fluorescence microscope (OLYMPUS, Tokyo, Japan). The cells were sub-cultured for further experiments when they reached 80–90% confluency.

### 2.3. Pro-Inflammatory M1-like Macrophage Induction and Co-Culture

THP-1 cells were purchased from BeNa Culture Collection (Xinyang, China) and cultured in RPMI 1640 medium (Gibco) containing 10% FBS at 37 °C in the presence of 5% CO_2_. Subsequently, 50 ng/mL PMA (Sigma) was used to induce THP-1 cells for 48 h. Then, 0.25 µg/mL lipopolysaccharide (LPS) and 20 ng/mL IFN-γ (Sigma) were added to the cells for another 48 h, thereby inducing differentiation into pro-inflammatory M1-like macrophages.

M1-like macrophages were co-cultured with HIF-MSCs (P3 generation) or MSCs (P5 generation) in Transwell permeable supports with a 0.4-μm polyester membrane (Corning International, Tokyo, Japan) for 48 h. Then, HIF-MSCs or MSCs were plated on the upper chamber in DMEM, and macrophages were plated on the lower chambers in RPMI 1640 medium.

### 2.4. Cell Phenotype and Viability

The expression levels of cell surface antigens (CD90, CD73, CD45, and CD34) on MSCs and HIF-MSCs and cell surface antigens (CD80, CD86, CD163, and CD206) on macrophages were detected using flow cytometry. Harvested cells were digested with trypsin-EDTA and stained with FITC anti-human CD34, APC anti-human CD73, PE anti-human CD90, PE anti-human CD45, PE anti-mouse CD86, PE anti-human CD163, FITC anti-human CD80 and FITC anti-human CD206 (Biolegend, San Diego, CA, USA), all human-specific antibodies. Immunofluorescence was performed to detect the expression of macrophage surface molecules F4/80, CD86, and CD163 in tissue sections. The expression levels were then analyzed using a BD FACS Calibur cytometer (Beckman, Pasadena, CA, USA). The viability of MSCs was evaluated using trypan blue exclusion and cell counting kit-8 (CCK-8) (Beyotime, Shanghai, China).

### 2.5. Colitis Mouse Model

Female BALB/c mice (6–8 weeks old) were obtained from Sipeifu Animal Technology Co. (Beijing, China). The mice were fed in a specific pathogen-free environment at 25 °C, exposed to a 12/12 h light/dark cycle, and were blindly sampled following a randomized design (six mice in each group). Colitis was induced using a 35% TNBS (Sigma) solution containing 20% anhydrous alcohol (Sigma). After stimulated defecation, mice were administered 0.1 mL of TNBS solution via rectal perfusion once every 7 days (four times in total) to imitate chronic colitis. Day 21 was the end point of modeling.

### 2.6. MSCs and HIF-MSC Transplantation

On D22, 0.1 mL of MSC (P5 generation) suspension (10^7^ cells/1 mL PBS), 0.1 mL of HIF-MSC (P3 generation) suspension (107 cells/1 mL PBS), or 0.1 mL of PBS was injected via the tail vein to the MSC, HIF-MSC, and TNBS groups 24 h after the last induction. The selected phosphatidylinositol 3-kinase-γ (PI3K-γ) inhibitor IPI549 (MCE, NJ, USA) was administered intraperitoneally and simultaneously with cell injection at a dose of 0.3 mg per animal.

D25 was the end point of observation. MSCs and HIF-MSCs colonized in intestinal tissue (Appendix A). Clinical assessment included weight, fur, stool, and activity status; the disease activity index (DAI) was scored based on weight loss, stool characteristics, and blood in the stool. The weight loss rate was determined as the loss in weight/primary weight over time. The pathological score was evaluated based on hematoxylin and eosin staining.

### 2.7. Macrophage Depletion

Clodronate liposomes (CL2MDP; Yeasen, Shanghai, China) were injected via the tail vein at 200 μL (5 mg/mL) per mouse every 2 days before and after HIF-MSC transplantation. Three days after injection, mice were sacrificed to collect serum and colon specimens. The CLP group treated the TNBS mice with CL2MDP, and the HIF-CLP group treated the TNBS mice with CL2MDP and HIF-MSC injection.

### 2.8. Inflammatory Cytokine and Target Gene Detection

ELISA kits (BIOFINE, Beijing, China) were used to detect inflammatory cytokines, including TGF-β, IL-10, IL-12β, and IL-17A, in the mouse serum. Colonic tissue fragments were lysed with RIPA lysis buffer (Beyotime, China), and the proteins were separated and transferred to PVDF membranes (Millipore, St, Louis, MO, USA).

Western blotting was performed to detect inflammatory cytokines and the proteins of target genes in colon tissue. Protein was extracted with an operation at 4 °C, as quickly as possible, to reduce the degradation of HIF-1α in normoxia, appropriately extending the splitting time. After being incubated with primary and secondary antibodies, the membranes were assessed using enhanced chemiluminescence (ECL). Densitometric analysis was performed using Image Pro-Plus (Media Cybernetics, Rockville, MD, USA).

Quantitative PCR (qPCR) was performed to analyze the expression levels of inflammatory cytokines and the target genes of cells. The primers used are listed in Appendix A.

### 2.9. Statistical Analysis

Quantitative data are expressed as the mean ± standard deviation. Statistical comparisons were performed using ANOVAs and *t*-tests in SPSS v25.0 (SPSS, Inc., Chicago, IL, USA). *p* < 0.05 was considered statistically significant.

## 3. Results

### 3.1. Cellular Morphology, Phenotype, and Viability of HIF-1α-Overexpressing MSCs

Isolated primary MSCs were long spindle-shaped cells distributed in a bundle or vortex with a closely arranged adherent growth, and the cell morphology did not change after passage or lentiviral transfection. Screened using puromycin, transfected GFP-expressing cells are defined as P1 HIF-MSCs. As depicted in Figure 1A, the intensity and density of the fluorescent signal in transfected cells did not decline with cell passage. As shown in Figure 1B, flow cytometry revealed that HIF-MSCs had a similar phenotype as MSCs; 92.6% and 98.7% of MSCs, as well as 92.3 and 98.7% of HIF-MSCs, were CD73+CD45- and CD90+CD34-, respectively. Cellular morphology and phenotype were not affected by lentiviral transfection.

The expression of HIF-1α was detected using qPCR and Western blotting. HIF-1α expression at mRNA and protein levels were remarkably elevated upon transfection and did not decline following cell passage (Figure 1C), indicating that HIF-1α overexpression via lentivirus transfection could effectively and stably upregulate HIF-1α expression in MSCs.

Growth of HIF-MSCs and MSCs transfected with the negative control lentivirus was both observed during the incubation, showing logarithmic growth and then a plateau, forming a typical S-type growth curve similar to untreated MSCs, demonstrating that neither lentivirus nor HIF-1α affected MSCs growth (Figure 1C). Trypan blue staining revealed that the viability of MSCs and HIF-MSCs was over 90%. CCK-8 showed that the optical density of HIF-MSCs increased significantly compared with MSCs at 1d, 2d, 3d, 4d, and 5d, indicating that cell viability was significantly enhanced (Figure 1E).

Qualitative PCR detection revealed that compared with initial MSCs, HIF-1α overexpression upregulated the expression of vascular endothelial growth factor (VEGF) as well as glycolysis-related indicators, including glyceraldehyde-3-phosphate dehydrogenase (GAPHD) and lactate dehydrogenase (LDHA). Likewise, the expression of the homing-related indicator CXCR-4 was upregulated (*p* < 0.05), suggesting a potential effect of HIF-1α overexpression on the function, energy metabolism, and homing of MSCs (Figure 1F).

### 3.2. HIF-MSCs Attenuate TNBS-Induced Experimental Colitis

Using a 35% TNBS enema solution, mice in the experimental model gradually experienced a decrease in vitality and body weight and exhibited anorexia, diarrhea, and bloody stools. The DAI was used to evaluate the general clinical status of the mice and was calculated according to weight loss, stool characteristics, and blood in the stool.

On day 21, at the end of molding, the weight loss rates of the model mouse group (TNBS group) were significantly higher than those of the normal control group. Following stem cell transplantation, the weight loss of the MSC and HIF-MSC group mice stopped after 3 days, while the weight of the TNBS group mice continued to decline (Figure 2A). On day 25, the weight loss rates of the HIF-MSC group were significantly lower than those of the TNBS and MSC groups (*p* < 0.05) (Figure 2B).

After stem cell transplantation, the DAI of the MSC and HIF-MSC groups decreased significantly compared with that of the TNBS group on day 25. Moreover, the mean value of the DAI of the HIF-MSC group mice was significantly lower than that of the MSC group mice (Figure 2C).

Compared with the control mice, the colonic mucosa of the model mice presented severe congestion, edema, bleeding, erosion, and ulceration; multiple segments of the bowel with alternating stenosis and expansion were observed. On day 25, 3 days after MSC and HIF-MSC transplantation, the degree of congestion, edema, bleeding, erosion, and ulceration was reduced. Compared with the TNBS and MSC groups, the shortened colon length recovered significantly in the HIF-MSC group (*p* < 0.05) (Figure 2D).

Histological examination was used to evaluate colitis further. High concentrations of neutrophils and lymphocytes were found in the submucosa, and abscesses (formed by the aggregation of inflammatory cells) were observed around the recess. On day 25, the histopathological damage of mice in the MSC and HIF-MSC groups was relieved, manifesting as the notable relief of congestion and edema, reduction of inflammatory cell infiltration, and partial repair of mucosal erosion. The pathological score was based on the depth of the lesion, the extent of recess destruction, the extent of the lesion, and the degree of inflammatory infiltration. The pathological score of the HIF-MSC group was significantly lower than those of the TNBS and MSC groups (*p* < 0.05) (Figure 2E,F).

In addition to local inflammation, the change in intestinal epithelial appearance suggested that HIF-MSC and MSC transplantation affected the epithelial integrity directly. There was a significant difference in the expression of intestinal tissue repair markers after transplantation in experimental colitis mice. The expression of Ki67, endothelial nitric oxide synthase (eNOS), and occludin was used to evaluate cell proliferation, colonic mucosal angiogenesis, and the integrity of the intestinal epithelium, respectively. Transplantation with MSCs and HIF-MSCs effectively elevated the expression of the three indicators. There was notably greater variation in expression levels of the three indicators in the HIF-MSC group than that in the MSC group (*p* < 0.05) (Figure 2G).

These results indicated that HIF-MSCs more effectively ameliorated intestinal inflammation symptoms than MSCs and prompted intestinal epithelium recovery.

### 3.3. HIF-MSCs Affect the Immune Balance of Mice and Colon Tissue via Macrophages

The levels of serum inflammatory factors in mice were detected using ELISA. The levels of anti-inflammatory factors, transforming growth factor-β (TGF-β) and interleukin 10 (IL-10), in the sera of the HIF-MSC and MSC groups mice increased, while those of pro-inflammatory factors, interleukin12β (IL-12β) and interleukin17A (IL-17A), decreased and were significantly different from those in the TNBS group mice. The increase in anti-inflammatory factors and the decrease in pro-inflammatory factors were more evident in the HIF-MSC group than in the MSC group. TGF-β and IL-17A levels were significantly different between the HIF-MSC and MSC groups (*p* < 0.05) (Figure 3A).

Western blotting was performed to detect tissue inflammatory factors. The change in inflammatory factor expression in the tissues was similar to that in the serum. Furthermore, the expression of colonic anti-inflammatory factors was higher and that of pro-inflammatory factors was lower in the HIF-MSC group than in the MSC group (*p* < 0.05) (Figure 3B).

To determine the role of macrophages in the immunoregulation of HIF-MSCs, clodronate liposomes (CL2MDP) were injected for the depletion of macrophages, including intestinal macrophages (CLP group). Immunofluorescence analysis revealed F4/80 expression clearly decreased more in the CLP group than in the TNBS group (Figure 3C). Compared with the TNBS group, HIF-MSCs did not alleviate the DAI and pathological score upon macrophage depletion (*p* > 0.05) (Figure 3D). Western blotting revealed that macrophage depletion notably reduced the influence of HIF-MSCs on inflammatory cytokines in the HIF-CLP-treated group, and there was no significant difference between the HIF-CLP-treated and TNBS groups (*p* > 0.05) (Figure 3E). Moreover, the differences between the CLP group and the TNBS group in DAI, pathological score, TGF-β, IL-10, and IL-17A were not significant (*p* > 0.05) (Figure 3D,E). The results demonstrated that the depletion of macrophages significantly impaired the benefits of HIF-MSC-treated colitis mice, thereby indicating the role of macrophages in the immunoregulation of HIF-MSCs.

### 3.4. HIF-MSCs Promote M1-like Macrophages Polarization toward M2-like Macrophages in Colitis Tissue

In vivo, immunofluorescence revealed that more GFP-labeled HIF-MSCs could colonize the intestine than GFP-labeled MSCs (Appendix A). To further investigate whether HIF-MSCs affect the polarization of macrophages by playing a role in immunoregulation, we analyzed the polarization of intestinal macrophages in colitis mice. Immunofluorescence analysis was used to detect F4/80, CD86, and CD163 in the colonic tissue. To minimize the impact of the difference in F4/80 expression, we calculated the relative ratio of merge expression and single F4/80 expression. The results demonstrated that HIF-MSC injection promoted a decrease in the relative expression ratio of F4/80+CD86+ and an increase in that relative expression ratio of F4/80+CD163+ in the colonic tissues of experimental colitis mice (*p* < 0.05) (Figure 4A). Moreover, Western blotting revealed that HIF-MSCs effectively upregulated M2-like macrophage-characteristic Arg-1 expression and downregulated M1-like macrophage-characteristic INOS expression in the colon tissue, thereby having a more significant effect than MSCs (Figure 4B).

### 3.5. HIF-MSCs Promote M1-like Macrophages Polarization toward M2-like Macrophages In Vitro

To investigate the effects of HIF-MSCs on inflammatory cells in vitro, we induced M1-like macrophages in THP-1 cells as a cellular immunological model (Group LPS) with LPS and IFN-γ. The expressions of inflammatory cytokines and macrophage-characteristic products were analyzed using qPCR. Co-culturing with MSCs and HIF-MSCs (Group M and H) decreased pro-inflammatory (TNF-α, IL-6, IL-12b, IL-23) cytokine expression and increased anti-inflammatory (TGF-β and IL-10) cytokine expression compared with induced M1 macrophages (Group LPS) and normal controls (*p* < 0.05) (Figure 5A). We also detected other characteristic products to explore the effects of MSCs and HIF-MSCs on macrophage subtype transformation. The expression of CCR-7 and INOS decreased significantly following co-culture with MSCs and HIF-MSCs, whereas the expression of Arg-1 increased significantly (*p* < 0.05). HIF-MSCs had a more significant effect on the regulation of inflammatory cytokines, Arg-1, CCR-7, and INOS expression than MSCs (*p* < 0.05) (Figure 5B).

Flow cytometry was used to detect the effect of HIF-MSCs and MSCs on induced M1-like macrophages, with CD80+ CD86+ cells representing M1-like macrophages and CD163+ CD206+ cells representing M2-like macrophages. The results demonstrated that co-culturing HIF-MSCs promoted a decrease in the proportions of M1-like macrophages and an increase in those of M2-like macrophages, being notably more effective than MSCs (*p* < 0.05) (Figure 5C).

### 3.6. HIF-MSCs Affect Macrophage Polarization through the PI3K-γ Pathway

In our previous study, we demonstrated that MSCs aggravate intestinal inflammation in PI3K-γ-knockout mice, suggesting the possible role of PI3K-γ in HIF-MSC therapy [25]. VEGF is an important factor secreted by MSCs and is widely recognized as an agonist of the PI3K signaling pathway. Consistent with mRNA expression, Western blotting revealed a significantly higher expression of VEGF in HIF-MSCs than in MSCs (Figure 6A). Subsequently, we detected the effect of HIF-MSCs on the downstream molecular targets of PI3K-γ, including HIF-1α and p-AKT/AKT, in the colon tissue. HIF-MSC treatment upregulated the expression of HIF-1α and p-AKT/AKT compared with that in the MSC treatment (*p* < 0.05) (Figure 6B,C).

To further elucidate the role of PI3K-γ in the effect of HIF-MSCs, a selected PI3K-γ inhibitor, IPI549, was used to block the PI3K-γ pathway, combined with HIF-MSC transplantation (Group HIF-PI3K-). Western blotting revealed that PI3K-γ inhibition blocked the upregulation effect of HIF-MSCs on HIF-1α and p-AKT/AKT in the HIF-PI3K(-) group (*p* < 0.05) (Figure 6B,C).

To verify how PI3K-γ influenced the immunoregulation of HIF-MSCs, we detected the inflammatory factors in the colon tissue of Group HIF-PI3Kγ- and found that the expression of IL-12β significantly increased, whereas that of the anti-inflammatory factors (TGF-β and IL-10) significantly decreased (*p* < 0.05) (Figure 6D).

Since PI3K-γ inhibition attenuated the effect of HIF-MSCs in increasing TGF-β and IL- 10 expression, we speculated that PI3K-γ inhibition might affect the influence of HIF- MSCs on macrophage polarization. We next focused on the effect of the PI3K-γ pathway in the regulation of macrophage polarization. Western blotting revealed that the expression of Arg-1 was notably decreased in Group HIF-PI3K(-), and the expression of INOS was increased (*p* < 0.05) (Figure 6E). Immunofluorescence analysis was used to detect F4/80, CD86, and CD163 expression in the colonic tissue of Group HIF-PI3K(-) (Figure 6F). Compared to the HIF-MSC group, PI3K-γ inhibition notably decreased the relative expression ratio of F4/80+CD163+ and increased the relative expression ratio of F4/80+CD86+ (*p* < 0.05), while there was no significant difference between the HIF-PI3K(-) and TNBS groups (*p* < 0.05) (Figure 6G). HIF-MSC injection promoted a decrease in the proportion of M1-type macrophages and an increase in that of M2 macrophages in the colonic tissues of experimental colitis mice, while PI3K-γ inhibition attenuated this effect.

To further elucidate the mechanism by which HIF-MSCs regulate macrophage polarization through the PI3K-γ pathway, we detected AKT subtype expression. Western blotting revealed that HIF-MSCs significantly upregulated AKT1 expression in the colonic tissues (*p* < 0.05) (Figure 6E). In vitro, IPI549, with a concentration of 20 nM, was used to inhibit the PI3K-γ effect in induced M1 macrophages. qPCR revealed that HIF-MSCs upregulated the expression of C/EBPβ and AKT1/AKT2 (*p* < 0.05), which promoted M1 macrophage polarization to M2 macrophages, as well as the expression of anti-inflammatory factors and M2 macrophage biomarkers. PI3K-γ inhibition blocked the effect of HIF-MSCs, leading to a significant decrease in the relative mRNA expression of Akt1/Akt2 and C/EBPβ (*p* < 0.05) (Figure 6H).

These results demonstrated the PI3K-γ pathway as a critical mechanism underlying the HIF-MSC-mediated promotion of macrophage polarization.

## 4. Discussion

MSCs are multipotent adult cells that have unique immunoregulatory properties that are currently being investigated as a treatment option for inflammatory disorders such as IBD. In our previous study, experimental colitis in mice was significantly improved after MSC transplantation. Thus, optimized therapeutic strategies targeting MSCs in IBD therapy are of great interest.

This study revealed three key findings: (1) through lentiviral transfection, HIF-1α-overexpressing MSCs were obtained in normoxic environments, with a significant increase in viability; (2) compared with MSCs, the expression of inflammatory cytokines was significantly affected by HIF-1α-overexpressing MSCs both in vitro and in vivo, thereby accelerating colonic mucosal repair in IBD experimental colitis mice; and (3) HIF-MSCs regulated immune balance by regulating the polarization of M1-like macrophages to M2-like macrophages, with relative anti-inflammatory effects, through a mechanism targeting the PI3k-γ pathway.

Hypoxia-sensitive pathways have an important impact on MSCs, and HIF-1α is critical for hypoxic adaptation. The advantages of MSCs cultured under hypoxic conditions include maintaining MSC proliferation, differentiation, metabolic balance, and other physiological processes [26]. As the hypoxic environment could not be maintained in vivo, we expected to increase the effectiveness of MSCs by intervening in the hypoxia-sensitive pathway. The modified MSCs we constructed maintained a high expression of HIF-1α at mRNA and protein levels without a decrease after passaging so that the characteristics obtained through genetic modification remained stable. In our study, HIF-1α overexpression in MSCs led to significantly enhanced viability and a rise in the mRNA expression of key glycolytic enzymes GAPHD and LDHA, which suggests an enhanced therapeutic effect of modified MSCs.

Herein, we have determined whether targeting related molecules in the hypoxic pathway of MSCs can be optimized as a treatment strategy, especially in a relatively hypoxic inflammatory intestinal environment. We established mouse models of chronic colitis, having inflammatory characteristics similar to those of repeated and progressive IBD, by repeated administration of low-concentration TNBS. We have demonstrated that both HIF-MSCs and MSCs could alleviate the severity of IBD, including clinical features and colon tissue pathology. The degree of remission exerted by HIF-MSCs was higher than that of MSCs, reflected in the improvement in DAI, body weight, colon length, and colon histology. The changes in the expression of Ki67, eNOS, and occludin led to the regeneration and repair of damaged tissues, which were all significantly higher in the HIF-MSC group. We, therefore, suggest that HIF-MSCs are more efficient in tissue repair and disease remission and suggest that HIF-MSCs can be regarded as an amplifier of the function of primary MSCs.

HIF-1α overexpression in MCSs influences cell-autonomous effects, including autonomous angiogenic and osteogenic effects [27]. The significantly enhanced cellular vitality and hypoxia adaptability help HIF-1α-overexpressing MSCs to have strong exocrine properties, immune regulation, and other functions. However, most studies involving HIF-affected MSCs in IBD have focused on the effects on tissue repair and angiogenesis, while our research was centered on immunological regulation through the intervention of HIF-1α overexpression in MSCs. We evaluated the immunoregulation upon MSC and HIF-MSC transplantation on the third day. We observed that HIF-MSCs more significantly affected the immune balance than MSCs, which was manifested in the colon rather than the whole body. It was reported that repair promoted by HIF-MSCs in intestinal mucosal inflamed tissues is closely associated with the immune response [28], which is consistent with our result that local immune balance reconstruction is closely associated with mucous rehabilitation.

The interaction with local immune cells is part of the complicated immune regulation effect of stem cells on immune-mediated disorders [29]. Several studies have proved that the HIF signaling pathway plays a role in the immunomodulatory effect of MSCs, including inducing higher expressions of IL6 in MSCs [30] and regulating Th17 and Treg differentiation, mediated by MSCs through the mTOR pathway [31]. It has been reported that HIF-1α overexpression modification enhances immunomodulation in dental MSCs [32] and improves the healing properties of extracellular vesicles by suppressing activated T-cells in Crohn’s disease [33]. Intestinal resident macrophages are at the front line of host defense at the mucosal barrier and are a potential therapeutic target in IBD [23]. Therefore, the intervention of macrophage polarization is an important requirement for immune regulation in the treatment of IBD.

The anti-inflammatory factors that we detected in the serum and tissue of colitis mice included TGF-β and IL-10, the two major cytokines secreted by alternatively activated macrophages (M2 macrophages) [34]. The changes in characteristic inflammatory factors caused by HIF-MSC transplantation led us to speculate that macrophages may be an important target of HIF-MSCs. Subsequently, we verified the role of macrophages in HIF-MSC therapy through macrophage depletion, and we confirmed that the immunoregulation effect of HIF-MSCs is highly dependent on macrophages. Immunofluorescence analysis results showed that MSC transplantation could regulate the M1/M2 ratio in inflammatory intestinal tissue, which confirmed our hypothesis. In vitro, HIF-MSCs similarly had an enhanced effect on macrophage polarization. Therefore, we concluded that HIF-MSCs promoting the polarization of M1-like macrophages into M2-like macrophages are one of the immunoregulating mechanisms in colitis mice.

The balance of pro-inflammatory M1 and anti-inflammatory M2 phenotypes is dynamic and strongly influences inflammation, which is critical for intestinal immune homeostasis [35]. Adoptive polarized M2 macrophages have protective effects on colitis mice [36]. Macrophage subtype predomination is temporally dynamic at different stages of inflammation, with a large spectrum of macrophage activation [37]. In addition, in the process of macrophage transformation, cell surface markers are changed. In our study, flow cytometry detected the cell expression of single CD80+ or single CD86+ and single CD163+ or single CD206+, suggesting the possibility of transitional morphology occurring in the co-culturing process. Moreover, the M2-like macrophages may be derived from the transformation of monocytes recruited from blood rather than only the intestinal M1-like macrophages [21]. Therefore, we only use the classical naming method, M1-like, or M2-like, to characterize whether macrophages have pro-inflammatory or anti-inflammatory effects and the detection of characteristic chemokines and enzymes of classic M1/M2 macrophages such as CCR7 [38], INOS, and ARG-1 to indicate the intestinal macrophages with different phenotypes. However, this change cannot represent any single intestinal macrophage, and further research is needed on primary intestinal macrophage extraction.

The PI3K-AKT pathway and its downstream targets have emerged as central regulators of the active phenotype in macrophages [39], and AKT1 and AKT2 protein kinases differentially contribute to macrophage polarization [40]. The effects of different isoforms of PI3K on macrophage activation are still controversial. It was reported that macrophages in PI3Kγ-deficient mice and humans had higher secretions of pro-inflammatory cytokines after pattern recognition receptor stimulation [41]. We showed that PI3K-γ attenuates the therapeutic effect of MSCs in IBD through PI3K-γ-knockout colitis mice. Kaneda et al. [42] reported the PI3Kγ-regulated immune suppression signature on M1 macrophages. Here, we observed the phenotype changes of M1-like macrophages induced by the PI3K-γ inhibitor in colitis tissue, suggesting that PI3K-γ inhibition significantly weakened the effects of HIF-MSCs in encouraging M2 macrophage polarization. The increased expression of TGF-β, IL-10, and IL-12b in the tissue through HIF-MSC transplantation was also attenuated upon PI3K-γ inhibition. In vitro, we demonstrated that HIF-MSCs promoted M1 polarization toward the M2 subtype through upregulated AKT1/AKT2 ratios, and C/BBPβ and PI3K-γ inhibition by IPI549 notably blocked this effect. The complexities of these results demonstrate that HIF-MSCs regulate macrophages through PI3K-γ, and upregulated VEGF in HIF-MSCs is a possible activator of the PI3K-γ pathway.

The PI3K/AKT/mTOR pathway has been demonstrated to primarily mediate non-hypoxic HIF regulation [43]. In our research, we also demonstrated that the selected PI3K-γ inhibitor attenuated HIF-MSC transplantation-induced HIF-1α overexpression in colonic tissues. HIF-1α overexpression in local inflammatory tissue promotes adaptive immunity by promoting lymph angiogenesis, and the TH1/TH17 response has a major impact on the development of inflammation [44]. We also found that the trend in HIF-1α expression in the colonic tissues was similar to that of inflammatory cytokines. Consequently, it may also be regulated by local inflammation rather than the merocrine secretions of homing HIF-MSCs.

HIF-1α was reported to be induced by Th1 cytokines in M1 macrophage polarization [45]; nevertheless, we observed that HIF-1α upregulation in colon tissue did not correspond with M1 polarization. This divergence may be because HIF-1α expression in colonic tissues could not represent it in specific macrophages as well as the influence of the HIF pathway on the other immune cells in IBD. In our study, we focused on immune regulation and the related mechanism exerted by MSCs with HIF-1α overexpression. The change in the HIF pathway caused by HIF-MSC transplantation and its influence on macrophages is another interesting research direction.

HIF-1α-overexpressing MSCs showed a superior immune regulation effect on colitis mice, but there is still insufficient research comparing this effect with that of primary MSCs. This may require further studies, such as RNA sequencing, to explore the specific mechanisms of HIF1α-overexpressing optimized MSCs in other aspects of therapy for colitis.

## 5. Conclusions

Herein, we have generated HIF-1α-overexpressing MSCs that show a stable overexpression in normoxic conditions using lentiviral vector transfection. HIF-1α-overexpressing MSCs effectively modulate the expression of inflammatory cytokines and M2-like macrophage polarization through PI3K-γ. Transplantation of HIF-1α-overexpressing MSCs in mice with experimental colitis resulted in good immune balance and mucosal rehabilitation, thereby proving a potentially effective treatment for IBD or other inflammatory diseases that can be applied in the future.

## Figures and Tables

**Figure 1 biomedicines-11-00825-f001:**
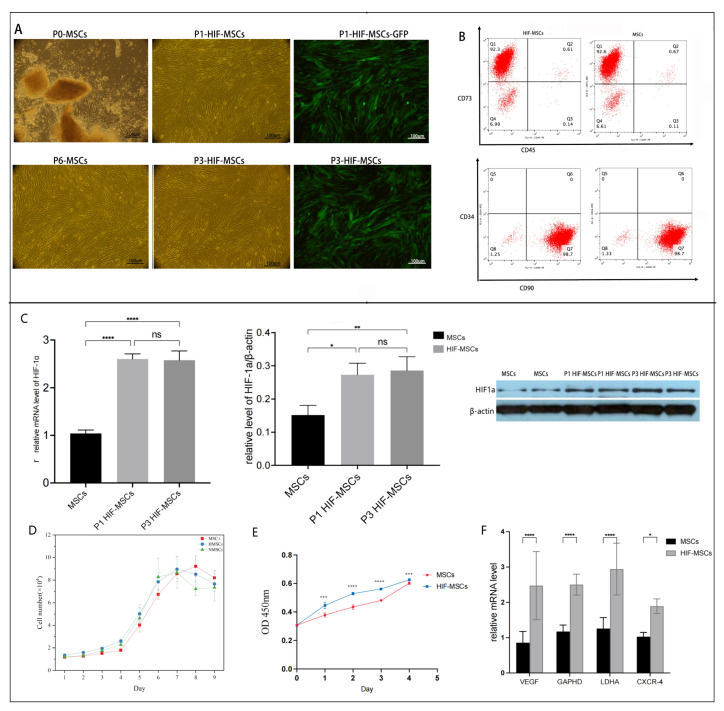
Features of HIF-1α-overexpressing MSCs (HIF-MSCs). (**A**). Morphology of MSCs before and after lentiviral transfection. (**B**). HIF-MSCs had a similar phenotype as MSCs. (**C**). Relative expression of HIF-1α in P1 and P3 HIF-MSCs at the mRNA and protein levels significantly increased compared to that in MSCs (**** *p* < 0.0001, ** *p* < 0.01, * *p* < 0.05). (**D**). HIF-MSCs had a similar typical S-type growth curve as MSCs. (**E**). Viability of MSCs and HIF-MSCs was assessed using CCK-8. HIF-MSCs had an enhanced viability compared to that of MSCs. (**** *p* < 0.0001, *** *p* < 0.001). (**F**). Relative expression of VEGF, GAPHD, LDHA, and CXCR-4 in HIF-MSCs at the mRNA level significantly increased compared to that in MSCs (**** *p* < 0.0001, * *p* < 0.05). “ns” represented no significant difference.

**Figure 2 biomedicines-11-00825-f002:**
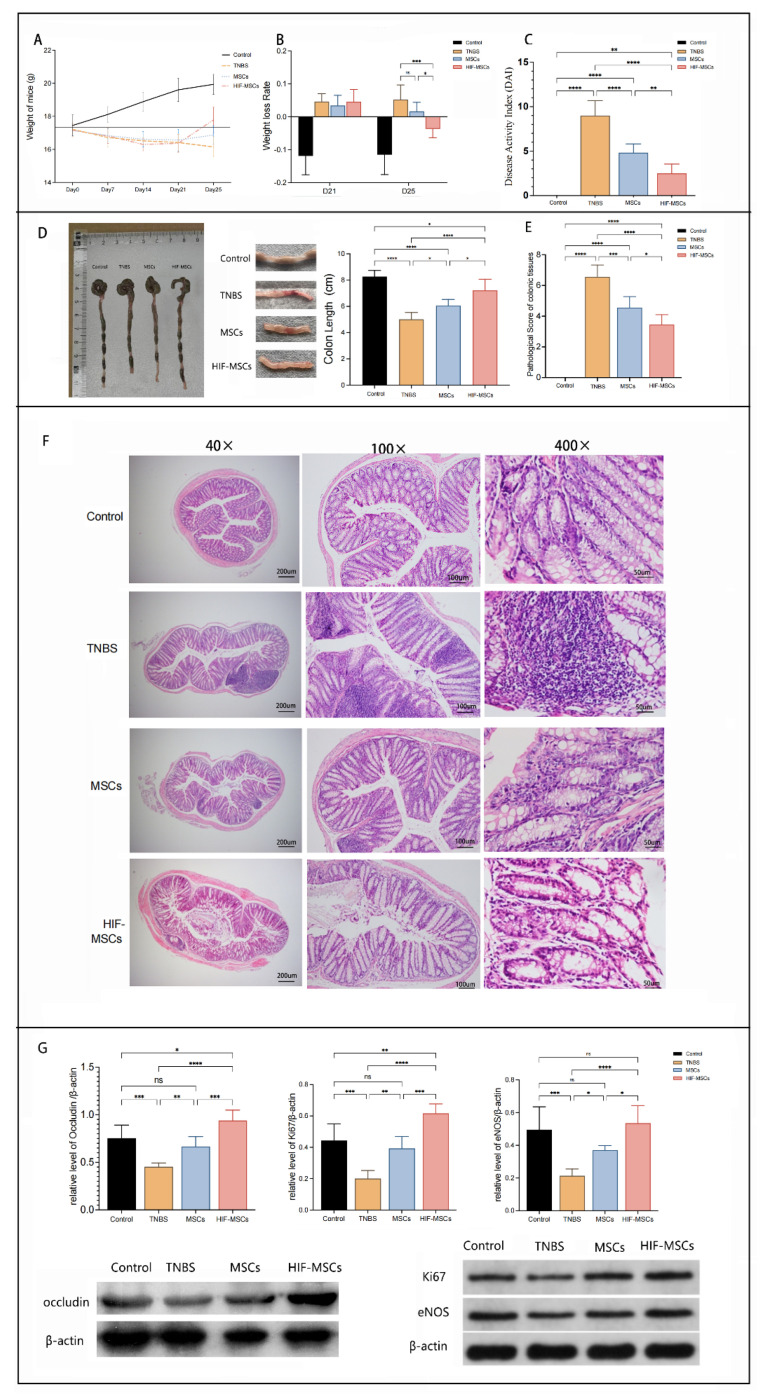
HIF−MSCs attenuate TNBS−induced experimental colitis. (**A**). Weight curve of mice. (**B**). Weight loss rate before and after transplantation (*** *p* < 0.001, * *p* < 0.05). (**C**). DAI decreased after HIF−MSC transplantation (**** *p* < 0.0001, ** *p* < 0.01). (**D**). The shortened colon recovered significantly in the HIF−MSC group compared with that in the MSCs group (**** *p* < 0.0001, * *p* < 0.05), and the degree of tissue lesion was obviously reduced (**E**). The pathology score of the HIF-MSC group was significantly decreased compared to those of the TNBS and MSC groups (**** *p* < 0.0001, *** *p* < 0.001, * *p* < 0.05). (**F**). Typical histopathological performance of colon tissue in each group. (**G)**. Expression of Ki67, eNOS, and occludin in the colon tissue was detected via Western blotting. Expression in the HIF−MSC group was significantly higher than those in the TNBS and MSC groups (**** *p* < 0.0001, *** *p* < 0.001, ** *p* < 0.01, * *p* < 0.05). “Ns” represented no significant difference.

**Figure 3 biomedicines-11-00825-f003:**
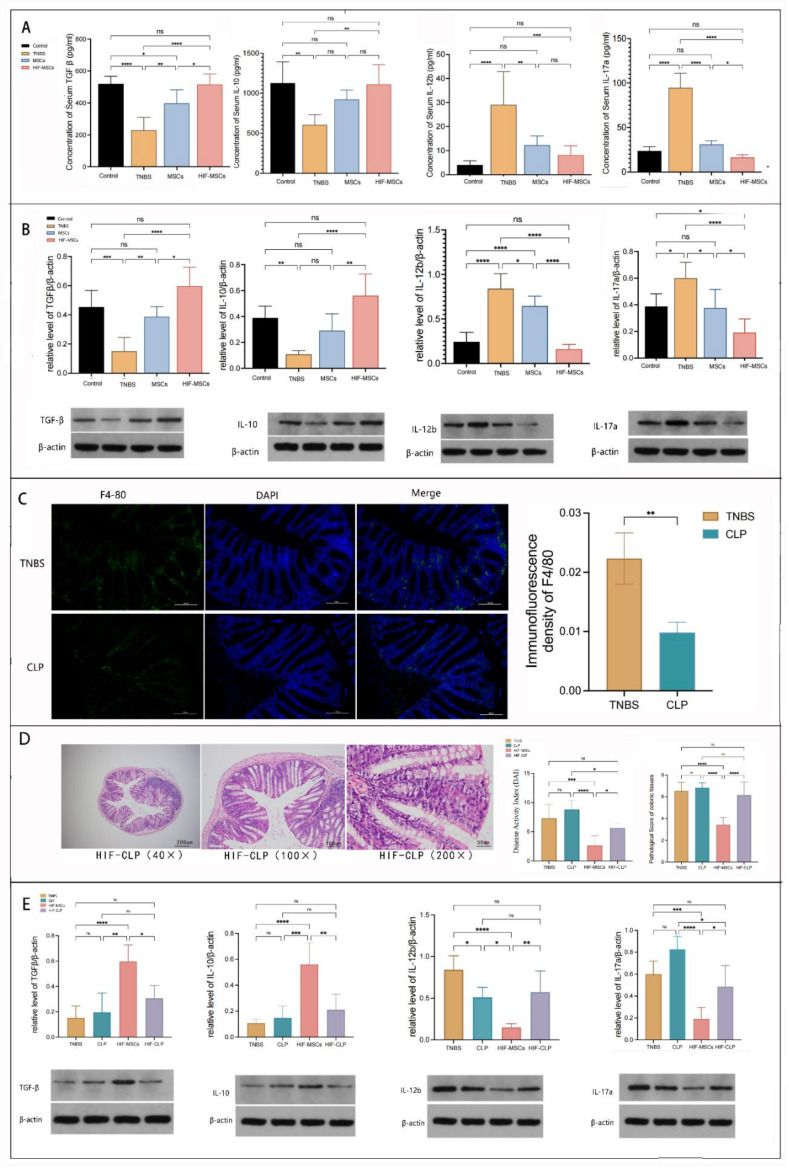
HIF-MSCs affect the immune balance of mice with colitis via macrophages. (**A**). HIF-MSCs increased the expression of anti-inflammatory factors TGF-β and IL-10 and decreased that of pro-inflammatory factors IL-12β and IL-17A in the serum (**** *p* < 0.0001, *** *p* < 0.001, ** *p* < 0.01, * *p* < 0.05). (**B**). HIF-MSC transplantation had a more noticeable effect on inflammatory cytokine expression in the colon tissue than MSC transplantation (**** *p* < 0.0001, *** *p* < 0.001, ** *p* < 0.01, * *p* < 0.05). (**C**). CL2MDP liposomes were intravenously injected 48 h before and after HIF-MSC transplantation (CL2MDP-HIF group). Single CL2MDP treatment with colitis mice was the CLP group. Immunofluorescence detected F4/80 expression with CL2MDP treatment; the expression in the CLP group was significantly lower than in the TNBS group, indicating effective intestinal macrophage deletion (** *p* < 0.01). (**D**). DAI and pathological performance were worsened in the CL2MDP-HIF group compared with indicators due to a single HIF-MSC treatment (**** *p* < 0.0001, *** *p* < 0.001, * *p* < 0.05). (**E**). CL2MDP attenuated the effect of HIF-MSCs on inflammatory factors (**** *p* < 0.0001, *** *p* < 0.001, ** *p* < 0.01, * *p* < 0.05). “ns” represented no significant difference.

**Figure 4 biomedicines-11-00825-f004:**
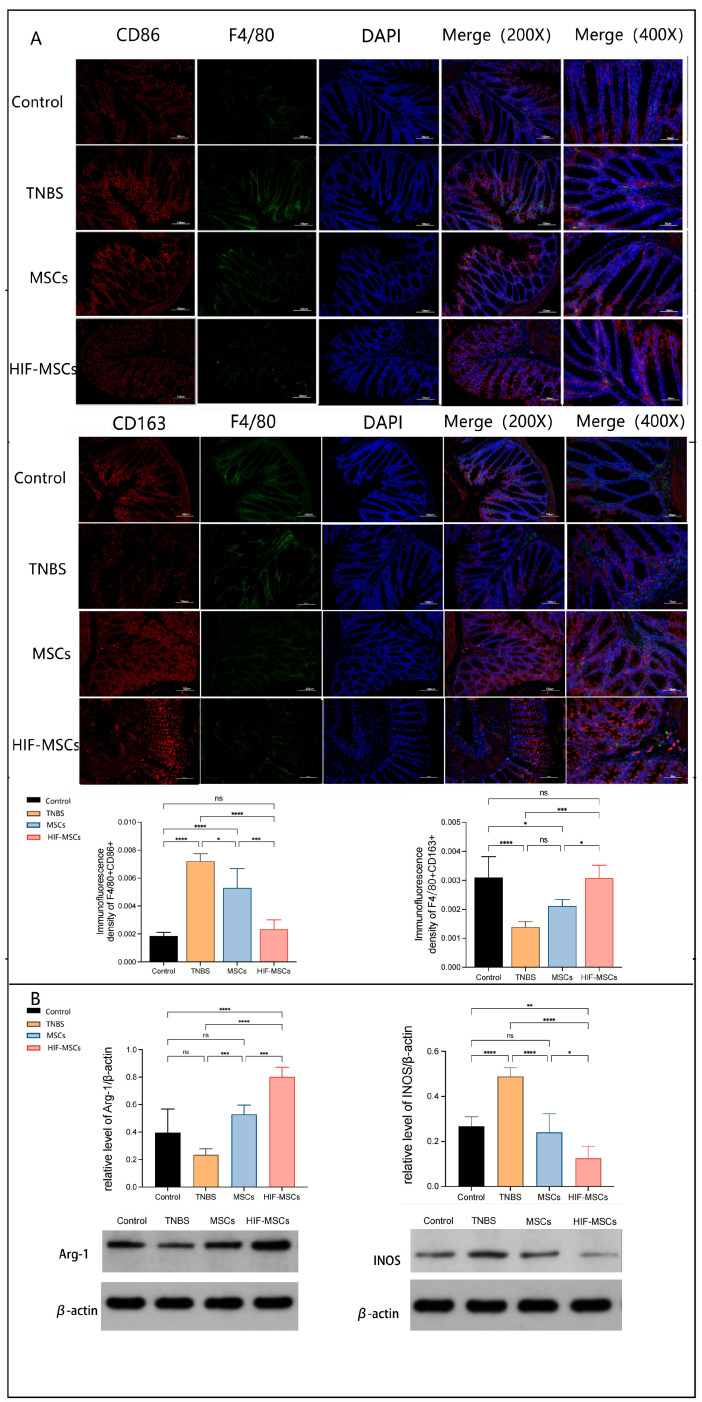
HIF-MSCs promoted M1 macrophage polarization toward M2 macrophages in vivo. (**A**). Immunofluorescence analysis was used to detect the expressions of F4/80+CD86+ and F4/80+CD163+ in colonic tissue, revealing that HIF-MSCs significantly decreased the relative expression ratio of F4/80+CD86+ and increased the relative expression ratio of F4/80+CD163+ compared with those promoted by PBS and MSCs (**** *p* < 0.0001, *** *p* < 0.001, * *p* < 0.05). (**B**). Western blotting analysis was used to detect the M2 characteristic Arg-1 and the M1 characteristic INOS. HIF-MSCs upregulated Arg-1 expression and downregulated INOS expression in colon tissue compared with PBS and MSCs (**** *p* < 0.0001, *** *p* < 0.01, ** *p* < 0.01, * *p* < 0.05). “ns” represents no significant difference.

**Figure 5 biomedicines-11-00825-f005:**
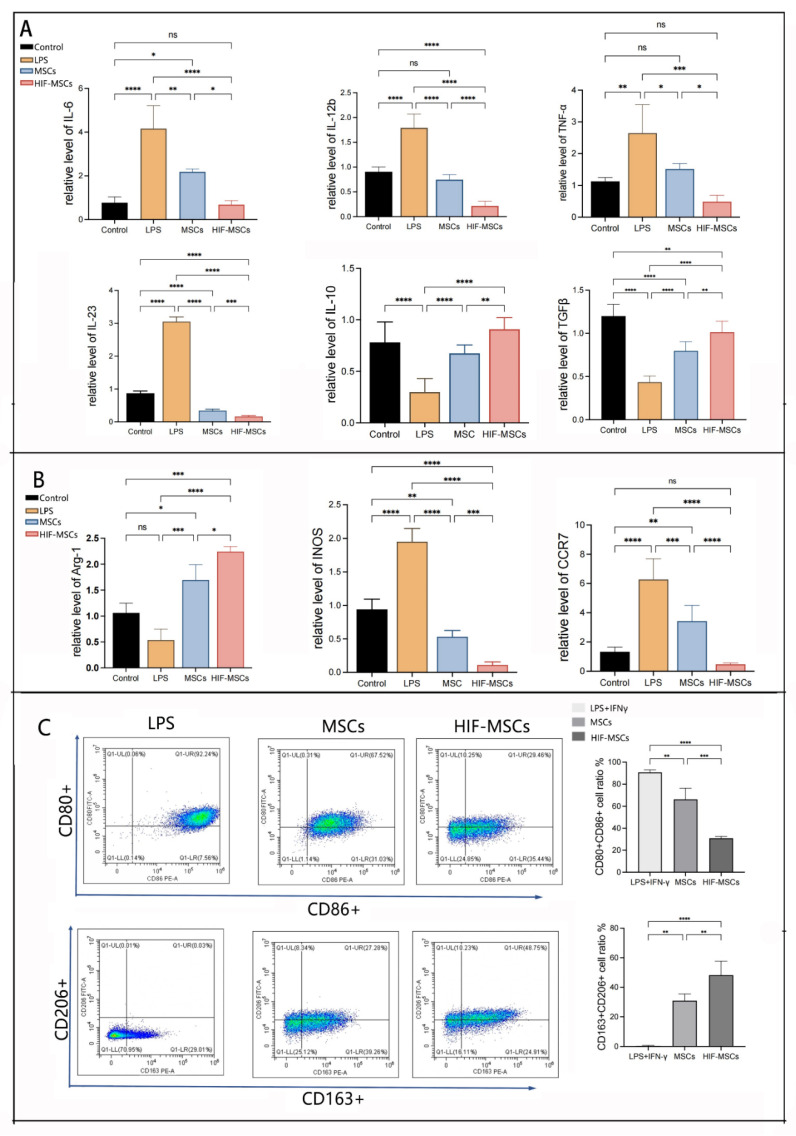
In vitro, PMA- and LPS/IFN-γ-induced M1-like macrophages were used as the cellular inflammation model, co-cultured with MSCs and HIF-MSCs for 48 h. qPCR was used to detect relative target gene expression in cells. (**A**). HIF-MSCs promoted anti-inflammatory cytokine expression and inhibited pro-inflammatory cytokine expression in M1 macrophages (**** *p* < 0.0001, *** *p* < 0.001). The difference in inflammatory cytokines expression was significant between HIF-MSCs and MSCs (**** *p* < 0.0001, *** *p* < 0.001, ** *p* < 0.01, * *p* < 0.05). (**B**). HIF-MSCs co-culturing effectively upregulated Arg-1 expression and downregulated INOS and CCR-7 expression in M1 macrophages (**** *p* < 0.0001, *** *p* < 0.001, ** *p* < 0.01, * *p* < 0.05). (**C**). Collected macrophages were labeled with anti-CD80, anti-CD86, anti-CD163, and anti-CD206 antibodies, then analyzed via flow cytometry in three independent experiments. M1-like macrophages were gated as double-positive CD 80+CD86+, and M2-like macrophages were gated as double-positive CD163+CD206. HIF-MSCs decreased the M1-like macrophage ratio and increased the M2-like macrophage ratio compared with those in the LPS and MSC groups (**** *p* < 0.0001, *** *p* < 0.001, ** *p* < 0.01). “ns” represented no significant difference.

**Figure 6 biomedicines-11-00825-f006:**
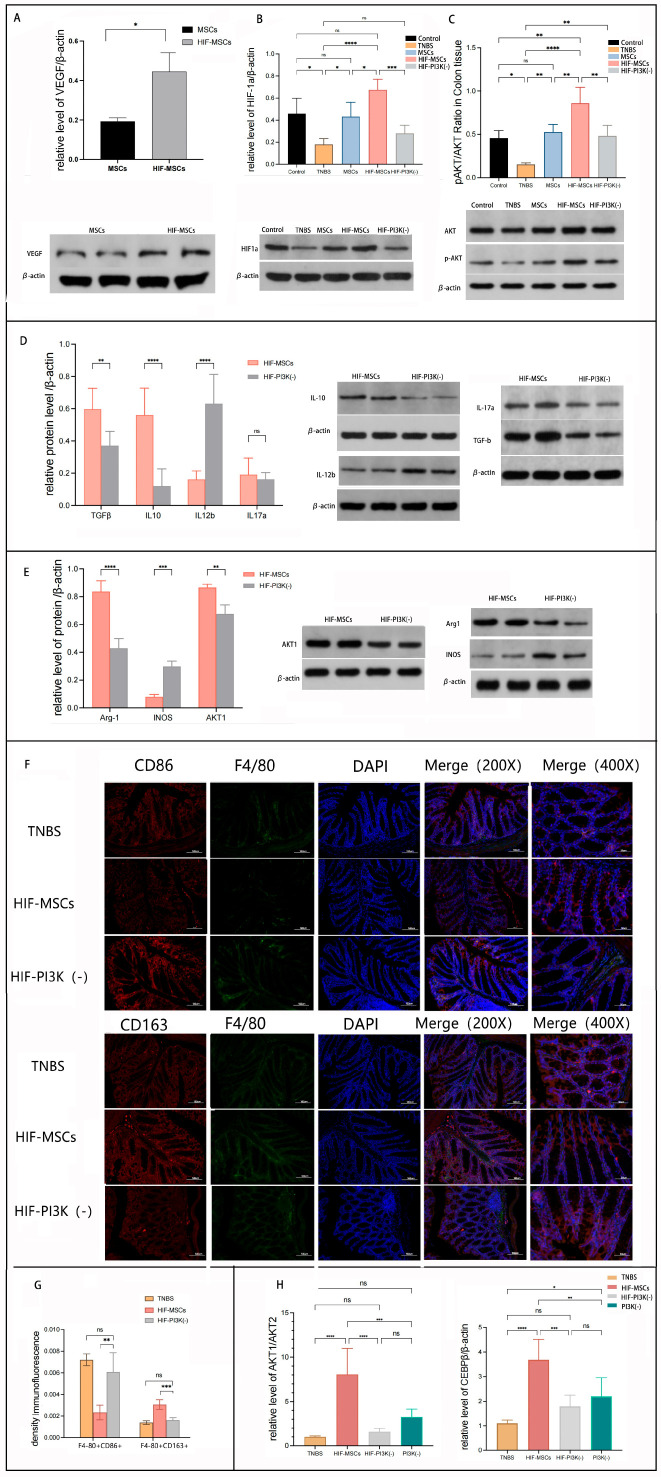
HIF-MSCs affected macrophage polarization through the PI3K-γ pathway. IPI549 was used to block the PI3K-γ pathway in mice and in induced M1-like macrophages. (**A**). Western blotting was used to detect VEGF in HIF-MSCs and MSCs. VEGF had a significantly higher expression in HIF-MSCs (* *p* < 0.05). (**B**,**C**). HIF-MSC treatment upregulated the expression of HIF-1α and *p*-AKT/AKT when compared with those of MSCs and PBS (**** *p* < 0.0001, *** *p* < 0.001, ** *p* < 0.01, * *p* < 0.05). PI3K-γ inhibition blocked the upregulatory effect of HIF-MSCs on HIF-1α and *p*-AKT/AKT (*** *p* < 0.001, ** *p* < 0.01). (**D**). PI3K-γ inhibition attenuated the regulatory effect of HIF-MSCs on inflammatory factors; there was a significant difference in IL-12b, TGF-β, and IL-10 expression between the HIF-MSC group and the HIF-MSC–PI3K-γ inhibition group (**** *p* < 0.0001, ** *p* < 0.01). (**E**). PI3K-γ inhibition decreased the expression of Arg-1 and AKT1 and increased that of INOS (**** *p* < 0.0001, *** *p* < 0.001, ** *p* < 0.01). (**F**,**G**). PI3K-γ inhibition decreased the relative expression ratio of F4/80+CD163+ and increased the relative expression ratio of F4/80+CD86+ (*** *p* < 0.001, ** *p*< 0.01) in colonic tissue compared with the HIF-MSC group. (**H**). qPCR was used to detect AKT1, AKT2, and C/EBPβ expression in macrophages. HIF-MSCs had upregulated AKT1/AKT2 and C/EBPβ expression; the effect was significantly blocked upon PI3K-γ initiation treatment (**** *p* < 0.0001, *** *p* < 0.001, ** *p* < 0.01, * *p* < 0.05). “ns” represents no significant difference.

## Data Availability

The raw data supporting the conclusions of this article will be available from the authors.

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
