# Peer review of "HIF-1α-Overexpressing Mesenchymal Stem Cells Attenuate Colitis by Regulating M1-like Macrophages Polarization toward M2-like Macrophages"

_biomedicines, 2023, doi:10.3390/biomedicines11030825_

Round 1

Reviewer 1 Report

Dear Authors,

The submitted manuscript entitled " HIF-1α Overexpression Mesenchymal Stem Cells Attenuate 2 Colitis by Regulating M1-like Macrophages Polarization to-3 ward M2-like Macrophages" is a well written and conducted study. Only minor recommendations i can suggest for this study.

1) Please describe in the introduction section, the basic characteristics of the MSCs, according to the latest literature.

2) Please briefly describe the role of HIF signaling pathway in MSCs immune mediated responses.

3) In the Materials and Methods section-2.1 MSC culture, please describe the explant method and the performed passages to obtain the required number of MSCs.

4) Did the authors performed the test for the minimum criteria for properly defying MSCs based on MSCs' committee of ISCT. Please include them as supplementary data.

Reviewer 2 Report

In this study, authors generated HIF-1alpha-overexpressing MSCs using lentiviral vector transfection technique.

Figure 4 and 5 may be revised to show clearer images. 

Author Response

Dear reviewer,

Thank you very much for your decision and constructive comments on our manuscript.

We have very carefully considered your suggestions and re-examined the images inserted in the manuscript. We found that the immunofluorescence pictures were not clear enough considering the picture’s size.

We have modified the resolution ratio of Figures 4 and 5 and revised Figures 4, 5, and 6 in the manuscript. If you have any further suggestions, please let us know and we will make further revisions. We have also prepared the original IF pictures for the graphic editor.

Reviewer 3 Report

In this article, entitled “HIF-1α Overexpression Mesenchymal Stem Cells Attenuate Colitis by Regulating M1-like Macrophages Polarization toward M2-like Macrophages”, the authors provide a potential therapeutic effect of mesenchymal stem cells for inflammatory bowel disease. Although some results are intriguing, there are several major comments.

1. HIF1a is unstable in normoxic conditions. HIF1a is only stable at O2 concentration below 5%. During normal cell culture conditions their half-life will be very short (~5min). The authors generated the HIF1a expressing MSCs but how can you convince this protein is working in normoxia? The authors suggest that the HIF1a will regulate the metabolism but cell growth does not show any significant changes (Fig1D). This could suggest that HIF1a is not working, although HIF-MSCs express more HIF1a than control MSCs.

2. The authors detect the changes of HIF1a with western blot. Usually detecting HIF1a via western blot is challenging because of the short half-life. I could not find a special method for sample preparation in the materials and Methods section.

3. After a few days of HIF-MSC injection, the pathology of TNBS-induced colitis is dramatically improved. How this improvement can happen within a short time? The authors inject MSCs via the tail vein. Can you check the MSCs in the colon? Since MSCs are derived from human samples, the authors can detect the MSCs by using human-specific antibodies. The authors need to show the existence of MSCs in the local area to convince the direct effect of MSC transplantation.

4. Usually, Clodronate depletes monocytes and macrophages only for a short term. Depleted cells quickly recovered within a few days. How could this temporal instinct depletion impact significant changes in colitis pathology (Figure 3)

5. Staining of macrophages does not make sense. F4/80 labels pan macrophages including both M1 and M2. When the author co-stain with CD86, expression of F4/80 is low in Control and HIF-MSC (Fig4A upper panel). However, when the author co-stain with CD163, the expression of F4/80 becomes opposite (high in Control and HIF-MSCs). This happens also in Figure 6F. F4/80 is low with CD86 in the HIF-MSCs (upper panel) but high with CD163 in the HIF-MSCs (lower panel). The expression of F4/80 should show always consistency regardless of co-staining. This suggests that the immunostaining data is not accurate.

Reviewer 4 Report

General Comments

The authors are genetically inducing the overproduction of HIF-1 genes in human MSCs and then infusing these and control cells into mice that have induced colitis.

It is an interesting project and important in chronic bowel disease is becoming a world-wide situation.

Specific Comments

P3,l116 THP-1 cells; what species?

You are using human specific antibodies and mouse specific antibodies and perhaps antibodies that are not species specific.  It is necessary in the text to indicate species specificity for each mention of an antibody.  For example, mCD90, hCD90 for mouse or human specific, CD90 for unknown specificity.

A problem with tail vein infusion of MSCs is that in many studies it has been shown that these cells do not freely circulate through the body.  Instead, they become entrapped in the lungs.  Consequently, in this study it is important to demonstrate the fate of infused human MSCs.  Do the infused cells make their way to the intestines?

This raises an important issue of whether the human MSCs are acting at a distance or are required to home to the site of the pathology.

Otherwise, the study indicates that the HIF-1 modified cells appear to exhibit a superior effect on reducing morbidity.

Round 2

Reviewer 3 Report

The authors addressed all questions well.

Author Response

Dear reviewer, 

Thank you very much for your careful review and valuable comments, which have greatly helped us to display the research results much better. Your suggestions also bring us enlightenment in the train of thought  and methodology. Sincerely appreciate  for your work!

Reviewer 4 Report

P1, l43;  Mesenchymal stem cells 43 (MSCs) are multipotent cells characterized with self-renewal potential,”  MSCs have limited self-renewal potential.  They age in culture upon repeated passaging.

Author Response

Dear reviewer,

Thank you for carefully reviewing the revised manuscript again. The expression "self-renewal potential" was  cited from the 6th literature. After careful consideration of your opinion, we discussed that this statement is indeed not rigorous enough, so we replaced it with "self-replication ability", to make the expression more accurate.

Your review comments have provided great help for better display of our manuscript. Thank you very much for your review!